# Strain-induced room-temperature ferroelectricity in SrTiO$_3$ membranes

Ruijuan Xu [1,2✉], Jiawei Huang[3], Edward S. Barnard [4], Seung Sae Hong[1,2], Prastuti Singh[1,2], Ed K. Wong[4], Thies Jansen[1], Varun Harbola[2,5], Jun Xiao [2,6], Bai Yang Wang[2,5], Sam Crossley[1,2], Di Lu[5], Shi Liu [3] & Harold Y. Hwang[1,2✉]

Advances in complex oxide heteroepitaxy have highlighted the enormous potential of utilizing strain engineering via lattice mismatch to control ferroelectricity in thin-film heterostructures. This approach, however, lacks the ability to produce large and continuously variable strain states, thus limiting the potential for designing and tuning the desired properties of ferro-electric films. Here, we observe and explore dynamic strain-induced ferroelectricity in SrTiO$_3$ by laminating freestanding oxide films onto a stretchable polymer substrate. Using a com-bination of scanning probe microscopy, optical second harmonic generation measurements, and atomistic modeling, we demonstrate robust room-temperature ferroelectricity in SrTiO$_3$ with 2.0% uniaxial tensile strain, corroborated by the notable features of 180° ferroelectric domains and an extrapolated transition temperature of 400 K. Our work reveals the enor-mous potential of employing oxide membranes to create and enhance ferroelectricity in environmentally benign lead-free oxides, which hold great promise for applications ranging from non-volatile memories and microwave electronics.

[1] Department of Applied Physics, Stanford University, Stanford, CA 94305, USA. [2] Stanford Institute for Materials and Energy Sciences, SLAC National Accelerator Laboratory, Menlo Park, CA 94025, USA. [3] School of Science, Westlake University, Hangzhou 310012 Zhejiang, China. [4] The Molecular Foundry, Lawrence Berkeley National Laboratory, 1 Cyclotron Road, Berkeley, CA 94720, USA. [5] Department of Physics, Stanford University, Stanford, CA 94305, USA. [6] Department of Materials Science and Engineering, Stanford University, Stanford, CA 94305, USA. ✉email: rxu3@stanford.edu; hyhwang@stanford.edu

Transition metal oxides exhibit a diverse set of electrical, magnetic, and thermal properties and hold great promise for modern technological applications. Among these oxide materials, the perovskite $SrTiO_3$ has stimulated considerable interest as it hosts a rich spectrum of physical properties such as dilute superconductivity[1], multiple structural instabilities[2], and a variety of emergent phenomena arising from the interface of $SrTiO_3$-based heterostructures[3–7]. In addition, $SrTiO_3$ is also one of the few known quantum paraelectric materials, in which quantum fluctuations and antiferrodistortive instabilities suppress ferroelectric polar order at low temperature, thus resulting in a nonpolar paraelectric state[8,9]. Despite the intrinsic paraelectric nature of $SrTiO_3$, it is possible to stabilize ferroelectric order via a variety of means such as substrate-induced strain[10–14], cation doping[15], $^{18}O$ isotope substitution[16], and defect engineering[17], etc. In particular, advances in thin-film epitaxy have highlighted the role of substrate-induced strain in stabilizing ferroelectricity and enhancing the ferroelectric transition temperature $T_c$ in $SrTiO_3$ thin-film heterostructures. This strategy of strain engineering relies on the lattice mismatch between the film and the underlying substrate, and has been widely used in tuning the structure and properties of many oxide materials[18–21]. However, due to the limited number of commercially available substrates and defect-induced strain relaxation during growth[22], this approach is fundamentally limited in its lack of ability to produce large and continuously tunable strain states. These constraints in turn limit the strain range that could be realized in practice, restricting the rational design, and control of desired material properties via strain.

The advent of freestanding crystalline oxide membrane films presents enticing possibilities to address these challenges and develop additional degrees of freedom to manipulate material properties. In particular, the recently developed water-soluble pseudoperovskite $Sr_3Al_2O_6$ has become widely used as a sacrificial buffer layer in the fabrication of a variety of freestanding, crystalline oxide thin films[23–28]. These freestanding films, with millimeter-scale lateral dimensions and down to nanometer-scale thickness, can accommodate much larger strains than their bulk counterparts[29–31]. Here, we integrate the freestanding $SrTiO_3$ films onto a flexible polymer stretching platform to probe the strain-tunable ferroelectric transition in $SrTiO_3$[30]. In this work, using a variety of characterization techniques we demonstrate robust room-temperature ferroelectricity in $SrTiO_3$ with 2.0% uniaxial tensile strain, which is corroborated by the notable features of 180° ferroelectric domains and an extrapolated transition temperature of 400 K.

## Results

**Fabrication of $SrTiO_3$ membranes.** First, we prepared an epitaxial heterostructure of 14 nm $SrTiO_3$ thin films with a 16 nm $Sr_2CaAl_2O_6$ sacrificial buffer layer synthesized on single-crystalline $SrTiO_3$ substrates via reflection high-energy electron diffraction (RHEED)-assisted pulsed-laser deposition (Supplementary Fig. 1). We substituted Ca into $Sr_3Al_2O_6$ to modify the lattice parameter of the sacrificial layer to closely match the $SrTiO_3$ lattice (the lattice constant of $Sr_2CaAl_2O_6$ is 15.6 Å, which is close to four times the lattice constant of $SrTiO_3$). Doing so can effectively reduce the lattice mismatch between different layers and minimize the crack formation in released freestanding films[27]. After fully dissolving $Sr_2CaAl_2O_6$ in deionized water, the $SrTiO_3$ film is released from the substrate and transferred onto a flexible polyimide sheet that can be stretched into various strain states (see "Methods" and Fig. 1a). The interface between the $SrTiO_3$ membrane and the polyimide sheet provides strong interface adhesion, enabling the success of the strain experiment.

Using both an optical microscope and atomic force microscopy, we characterized the topography of the resulting freestanding membranes and noted that the millimeter-scale films (laterally) are free of cracks in their unstrained state (Fig. 1b–d). Since strain relaxation occurs in the vicinity of cracks[32], these crack-free $SrTiO_3$ membranes are able to preserve homogeneous strain, presenting an ideal platform for our strain experiments. In addition, a two-dimensional array of gold electrodes was evaporated on the membrane surface using electron-beam evaporation with a shadow mask. These electrodes serve as optical markers to measure strain in the strain experiment (Fig. 1c).

**Characterization of room-temperature ferroelectricity.** Next, we conducted piezoresponse force microscopy (PFM) to explore how strain affects ferroelectric order in $SrTiO_3$ membranes. Here, the flexible polyimide sheet allows the strain state to be flexibly manipulated. In this work, we focus on the case of in-plane uniaxial tensile strain by stretching the membrane along the [100] direction, while supplying a small amount of tensile stress to keep the membrane undeformed along the other orthogonal in-plane direction. Note that it is usually difficult to create such a highly anisotropic strain geometry using commercially available substrates. The strain state was characterized optically by measuring the change in spacing between gold markers (i.e. $\varepsilon = \Delta l/l$, Fig. 2a), and further microscopically confirmed by grazing incidence X-ray diffraction (GIXRD) measurements[30] (see "Methods" and Fig. 2b–d). Here, with increasing strain, the GIXRD peak measured along the [100] strain direction shifts towards lower angles, indicating the increase in the lattice parameter upon stretching, whereas the peak position measured along the [010] direction remains almost unchanged. It is also noted that the uniaxial strain values measured by GIXRD closely match with the optically measured strain values. In order to maintain the strain state of the film even after removing the external stress from membranes, the adhesive polycaprolactone was used in its melted liquid form to bond the stretched membrane onto a rigid substrate at 110 °C, and then cooled to room temperature to lock-in the strain state (see "Methods"). Using this strain setup, we characterize the ferroelectric properties of strained $SrTiO_3$ via PFM at room temperature (see "Methods"). In a small strain state ($\varepsilon \leq 1.25\%$), we measured very weak signals from the membrane, indicating the absence of ferroelectricity at room temperature within this strain range (Fig. 2e–g and Supplementary Fig. 2). By contrast, for larger strain ($\varepsilon > 1.5\%$), we observed strong lateral signals from membranes with notable stripe domain patterns, which indicates the emergence of room-temperature ferroelectricity (Fig. 2h, i and Supplementary Fig. 2). The observed polydomain structures, which are only observable from lateral piezoresponse, are in-plane polarized due to the tensile strain.

Since lateral PFM imaging is carried out via the torsional movement of the PFM cantilever in response to shear deformation of in-plane polarized domains, in-plane piezoresponse signals will vanish when the cantilever is aligned along the in-plane polarization direction. Therefore, by varying the relative orientation between the cantilever and the sample, we can determine the actual polarization direction (Supplementary Fig. 3). Using this approach, we found that the ferroelectric polarization of $SrTiO_3$ membranes is along $[100]/[\bar{1}00]$ with the adjacent domains polarized at a 180° difference, which coincides with the uniaxial tensile strain direction (Fig. 2j). These PFM results provide direct evidence of robust room-temperature ferroelectricity in strained $SrTiO_3$ membranes, corroborated by the notable 180° ferroelectric polydomain structures. Moreover, we find that the $SrTiO_3$ membranes can sustain beyond 2.0% uniaxial tensile strain without fracture, which exceeds the

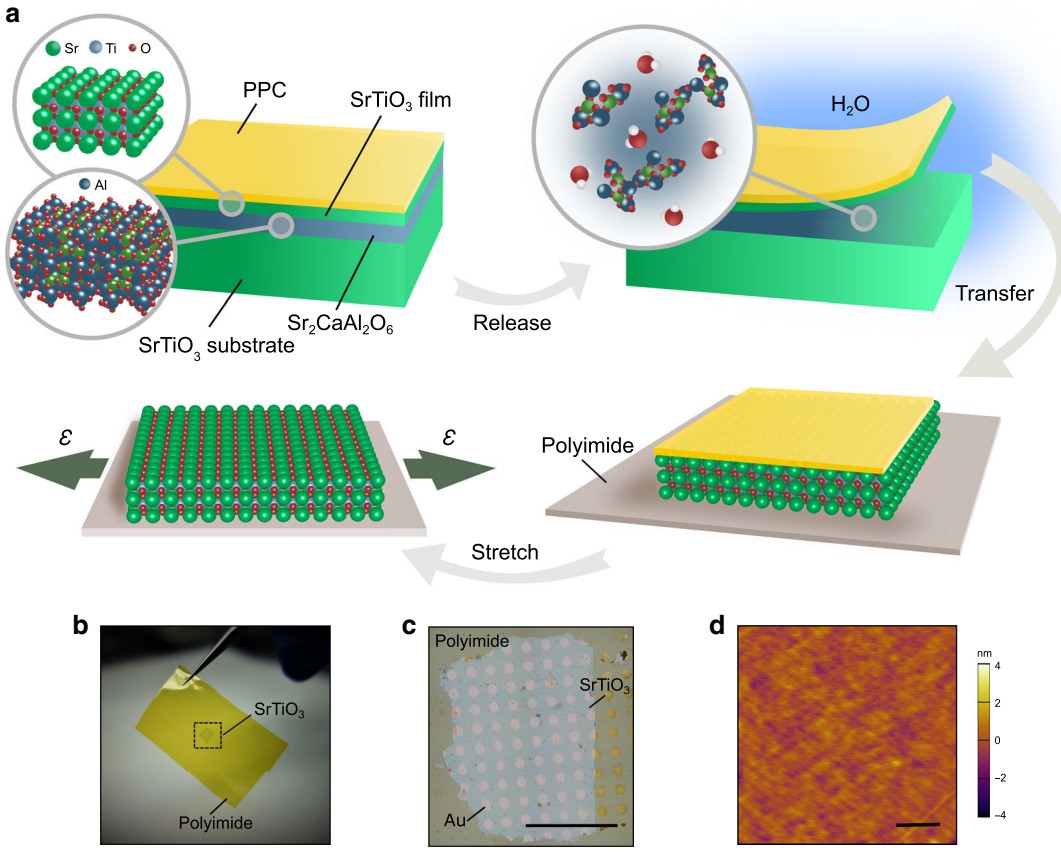

**Fig. 1 Preparation of freestanding SrTiO₃ membranes. a** Schematic illustrating the lift-off and transfer process for SrTiO₃ membranes onto polyimide sheets. By dissolving the sacrificial layer Sr₂CaAl₂O₆ from the as-grown heterostructure, SrTiO₃ films can be released and transferred onto a polyimide sheet with a layer of polypropylene carbonate (PPC) as the supporting material. The oxide/polymer bilayer structure can be stretched after the PPC is thermally decomposed in O₂. Optical images of the transferred millimeter-scale freestanding SrTiO₃ membranes are shown in **b** and **c** with an ordered array of gold as optical markers. The scale bar is 1 mm. **d** Detailed atomic force microscopy topographic images reveal the crack-free surface of SrTiO₃ transferred onto the polyimide sheet. The scale bar is 1 μm.

reported maximum substrate-induced tensile strain in SrTiO₃ heterostructures[10] (i.e., 1.0% for SrTiO₃ films grown on DyScO₃ (110)). Abrupt crack formation occurs typically above 2.5% strain in SrTiO₃ membranes, giving rise to paraelectricity in SrTiO₃ at room temperature due to strain relaxation (Supplementary Fig. 4).

**Optical second harmonic generation measurements.** We further performed temperature-dependent optical second harmonic generation (SHG) measurements to explore the strain-induced variation of $T_c$ in the SrTiO₃ membranes. In our SHG measurements, a 900 nm fundamental beam is used to excite the SHG signal from the membrane in a reflection geometry that can be probed at a wavelength of 450 nm (see "Methods" and Fig. 3a). First, in order to understand the structural symmetry of the strain-induced ferroelectric phase, we carried out SHG measurements in strained membranes that exhibit room-temperature ferroelectricity ($\varepsilon \geq 1.5\%$) as a function of incident beam polarization (Supplementary Fig. 5). The intensity of the output SHG signals is detected at a polarizer angle which is either parallel ($I_x$, Fig. 3b) or perpendicular ($I_y$, Fig. 3b) to the uniaxial strain direction in membranes. By analyzing these polar plots using the symmetry-based SHG tensor, we find that the ferroelectric phase is in the orthorhombic $mm2$ point group symmetry with the polar axis aligned in-plane along the uniaxial strain direction[33] (see "Methods"), which is consistent with our PFM observations. Next, we measured the ferroelectric $T_c$ of strained membranes by

probing SHG as a function of temperature. Since the adhesive used in our strain setup softens and allows strain relaxation above 60 °C, our measurements were limited to temperatures below this scale. We probed the $T_c$ of membranes strained to $\varepsilon = 0.5\%$, 0.9%, and 1.25%, wherein the SHG intensity decreases with temperature and gradually vanishes at a critical temperature which indicates the onset of the phase transition (Fig. 3c and Supplementary Fig. 6). Plotting the integrated SHG peak intensity as a function of temperature for each strain state, we can extract $T_c$ using a temperature-dependent order parameter fit derived from the Ginsburg–Landau–Devonshire (GLD) model (see "Methods" and Supplementary Fig. 7). Our results reveal that the measured $T_c$ increases linearly with strain, which agrees well with the theoretical value predicted by the GLD model (see "Methods", Fig. 3d and Supplementary Fig. 8). Following this theoretical trend, we can also estimate $T_c$ for membranes with a phase transition far above room temperature ($\varepsilon \geq 1.5\%$, Fig. 3d). Our results indicate that for 2.0%, $T_c$ extrapolates to 400 K, i.e. robust room-temperature ferroelectricity. In addition, these results also indicate that it is possible to directly tune the transition temperature in a deterministic manner.

**First-principles calculations and MD simulations.** To understand the origin of the strain-driven ferroelectric phase and the nature of the phase transition in strained SrTiO₃ membranes, we performed first-principles density functional theory (DFT) calculations and molecular dynamics (MD) simulations. We

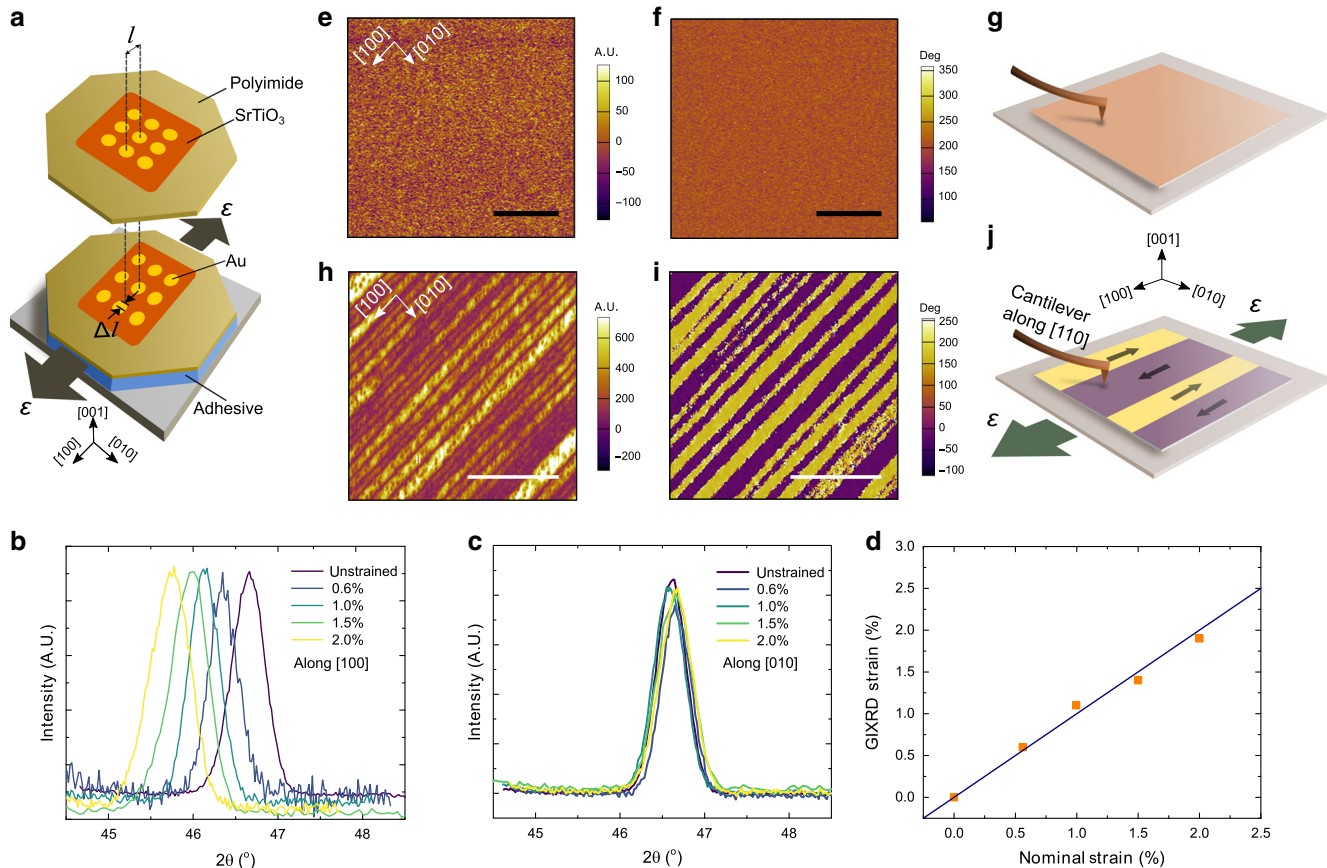

**Fig. 2 Characterization of room-temperature ferroelectricity in strained SrTiO₃ membranes. a** Schematic illustrating the strain apparatus, wherein the SrTiO₃ membrane and polyimide sheet are stretched and fixed to a rigid substrate to maintain their strain state, with the ordered array of gold markers used for strain calibration. Grazing incidence X-ray diffraction (GIXRD) results measured along **b** [100] strain direction and **c** [010] direction which was fixed to the unstrained state. **d** Comparison of GIXRD strain and the optically measured nominal strain. Error bars within the size of the marker represent the standard deviation. Piezoresponse force microscopy (PFM) amplitude **e** and phase **f** measured from unstrained membranes exhibit weak piezoresponse, indicating the absence of room-temperature ferroelectricity, illustrated in schematic **g**. PFM amplitude **h** and phase **i** measured from 2.0% uniaxially strained membranes exhibit room-temperature ferroelectricity with the notable 180° domain structure, illustrated in schematic **j**. The scale bar is 1 μm.

calculated the energy of the paraelectric and ferroelectric phase in SrTiO₃ as a function of strain using DFT calculations with a 5-atom unit cell and local density approximation (LDA) (see "Methods" and Fig. 4a). DFT calculations reveal that the ferroelectric phase is favored over the paraelectric phase by a small energy difference when the applied strain is >0.25%. Also, consistent with our experimental observations, DFT calculations indicate that uniaxial tensile strain along the [100] direction induces polarization along this direction, and the induced polarization is a result of the displacement of both the Sr and Ti atoms away from the center of the surrounding oxygen lattice (Supplementary Fig. 9, Table 1, and Supplementary Data 1). Next, slab-model MD simulations were performed to understand the nature of temperature-driven phase transitions in 2.0% uniaxially strained SrTiO₃ membranes. In order to obtain atomistic insights into the nature of the phase transition, we calculated the probability distribution of the unit cells adopting a [100]-component of local polarization as a function of temperature (Fig. 4b). At low temperature, the distribution of local polarization in the ferroelectric phase is Gaussian-like with a single peak at ≈0.3 C m⁻² (120 K, Fig. 4b). With increasing temperature, the peak shifts towards a lower polarization value, indicating the displacive character of the phase transition (e.g., 140 and 160 K, Fig. 4b). At 180 K, another peak located near −0.18 C m⁻² emerges, suggesting the onset of an order-disorder phase transition (180 K,

Fig. 4b). In the high temperature paraelectric phase (240 and 300 K, Fig. 4b), the distribution becomes a double-peaked curve (again an indicator of the order-disorder transition) but with a non-zero probability at $P = 0$ (an indicator of the displacive transition)[34], indicating a notable mixture of displacive and order-disorder characteristics of the phase transition in SrTiO₃ (Supplementary Fig. 10). Snapshots of the dipole configuration from MD simulations further illustrate such mixed transition character (Fig. 4c). For instance, the snapshot obtained at 120 K shows a typical ferroelectric phase with the majority of unit cells polarized along [100]. As the temperature increases, the orientation of the dipoles become more disordered but the long-range correlation still remains along [100]. At 300 K the macroscopic paraelectricity arises as an ensemble-averaged result of randomly oriented local dipoles[17], including both the nearly zero and non-zero dipoles, resulting in a zero-net polarization.

## Discussion

We observe direct evidence of robust room-temperature ferroelectricity in strained SrTiO₃ membranes, corroborated by 180° domain formation and the evolution of $T_c$ with strain. Using SrTiO₃ membranes as a promising example, our work demonstrates the significant potential of employing oxide membranes for enhanced ferroelectric properties in diverse oxide materials. These significant enhancements in ferroelectric polarization and

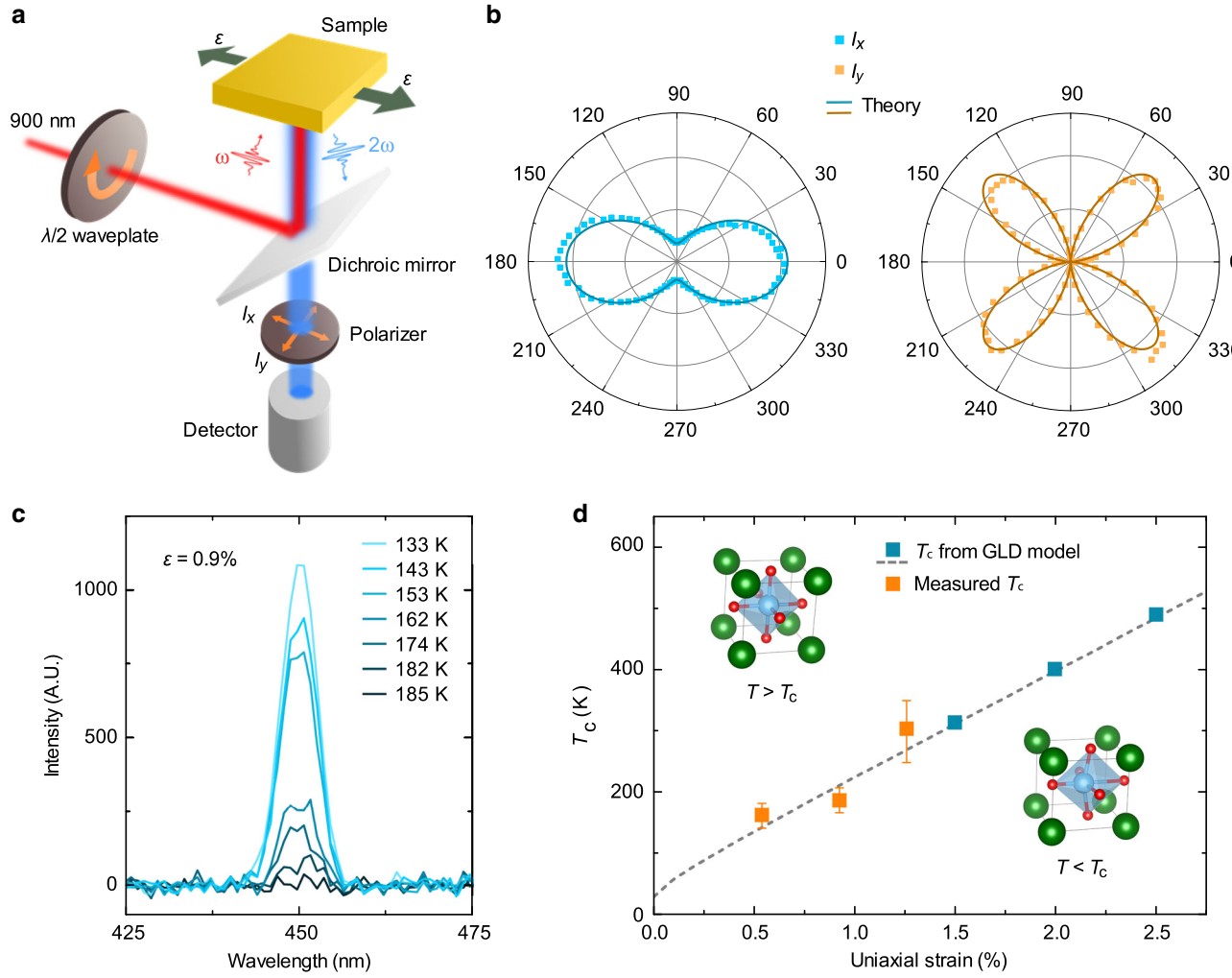

**Fig. 3 Optical second harmonic generation (SHG) measurements of SrTiO₃ membranes. a** Schematic illustrating the experimental setup for SHG measurements, wherein a 900 nm fundamental beam is used to excite the frequency-doubled signal from membranes in a reflection geometry that can be probed at a wavelength of 450 nm. **b** SHG polar plots measured at room temperature as a function of incident beam polarization in membranes strained at 2.0%. **c** SHG signals measured as a function of temperature in membranes strained at 0.9%. **d** $T_c$ plotted as a function of uniaxial strain directly measured from the SHG measurements, as well as high temperature extrapolations of the Ginsburg–Landau–Devonshire model. Error bars represent the standard deviation.

$T_c$ hold great promise for nonvolatile ferroelectric memory applications. The ability to obtain extreme and continuously tunable strain states with freestanding membranes also provides broad opportunities for achieving high dielectric turnability at room temperature, which is important for microwave electronics such as tunable capacitors and phase shifters, etc[35]. In addition, combining the nanoscale freestanding film with the flexible polymer substrate allows the strain state to be manipulated to arbitrary geometries and anisotropies, providing additional degrees of freedom for creating unconventional polar structures and functional domain walls coupled with potentially large dielectric, piezoelectric, and magnetic responses[36,37]. For instance, the observed in-plane polarized 180° domain structures in our work with only one in-plane polarization variant (along the [100]/[$\bar{1}$00] directions) is rather rare not just in SrTiO₃ but also in other ferroelectric perovskites such as PbTiO₃, BaTiO₃, etc. Such in-plane polarized domain structures could become potential candidates for device elements in next-generation nanoelectronics such as in-plane ferroelectric nonvolatile memories. Moreover, the polymer substrates which are stretched in the elastic deformation regime allows the reversible control of the phase transition and their associated dielectric and piezoelectric

responses, providing possibilities for designing a variety of strain-tunable ferroelectric devices.

## Methods

**Thin-film growth.** The epitaxial heterostructure of 14 nm SrTiO₃ films was synthesized with a 16 nm Sr₂CaAl₂O₆ sacrificial layer on (001)-oriented single-crystalline SrTiO₃ substrates via RHEED-assisted pulsed-laser deposition. The growth of the Sr₂CaAl₂O₆ layer was carried out in dynamic argon pressure of $4 \times 10^{-6}$ Torr, at a growth temperature of 710 °C, a laser fluence of 1.35 J cm$^{-2}$, and a repetition rate of 1 Hz, using a 4.8 mm² imaged laser spot. The growth of the SrTiO₃ layer was conducted in dynamic oxygen pressure of $4 \times 10^{-6}$ Torr, at a growth temperature of 710 °C, a laser fluence of 0.9 J cm$^{-2}$, and a repetition rate of 1 Hz, using a 3.0 mm² imaged laser spot.

**SrTiO₃ membrane fabrication.** The heterostructure was first attached to a polymer support of 100-μm-thick polypropylene carbonate (PPC) film and placed in deionized water at room temperature until the sacrificial Sr₂CaAl₂O₆ layer was fully dissolved. The PPC coated SrTiO₃ film was then released from the substrate and transferred onto a polyimide sheet. Finally, the PPC layer was removed from the membrane through thermal decomposition in O₂ at 260 °C for 2 h.

**Stretching experiment.** We applied uniaxial stress to SrTiO₃ membranes along [100] directions via micromanipulators to stretch the polyimide substrate, while keeping the membrane undeformed along the other orthogonal in-plane direction by applying a small amount of compensating stress along [010]. The resultant

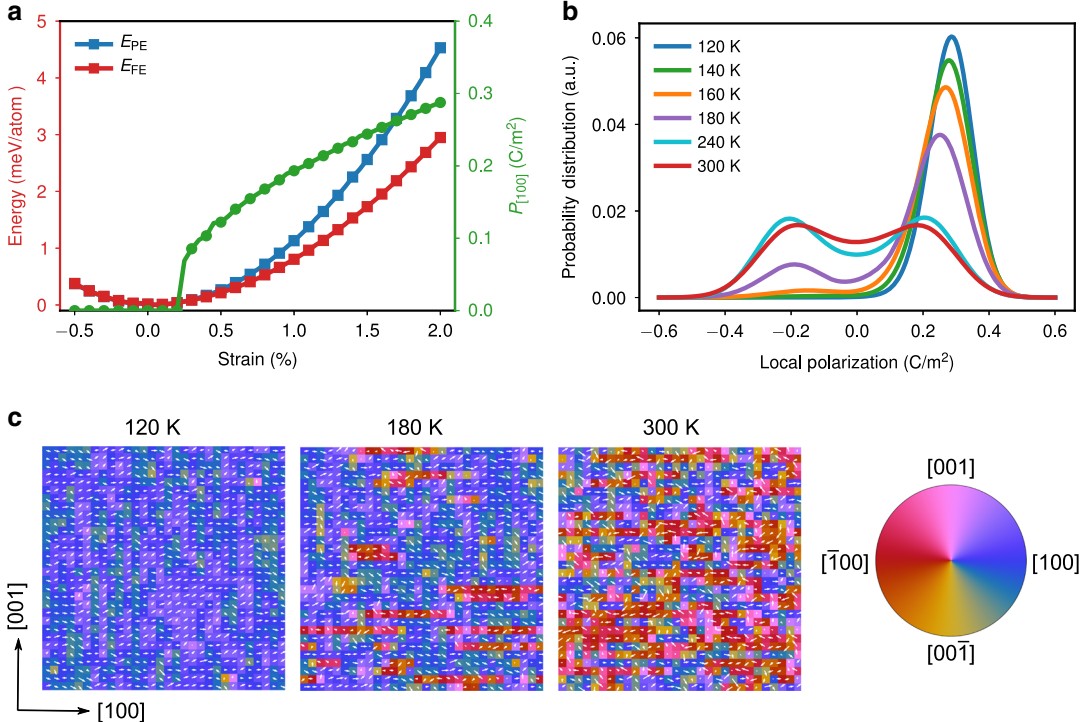

**Fig. 4 Density functional theory (DFT) calculations and molecular dynamics (MD) simulations. a** Calculated energy of paraelectric and ferroelectric phases and the induced ferroelectric polarization as a function of [100] uniaxial strain in SrTiO₃ from DFT calculations. **b** Probability distributions of the unit cells adopting a [100]-component of local polarization at various temperatures. **c** Snapshots of the dipole configurations at different temperatures obtained from MD simulations under 2.0% uniaxial strain conditions. Each white arrow in these graphs represents the local electric dipole within a unit cell, with a background color illustrating the polarization direction.

strain was characterized using optical microscopy to measure the change in spacing between circular gold markers, which were evaporated with a shadow mask using e-beam evaporation. In order to preserve the strain state in SrTiO₃, poly-caprolactone was used in its melted liquid form to bond the strained oxide/polymer bilayer to a ceramic chip carrier at 110 °C. Polycaprolactone solidifies by cooling the strain setup to room temperature, which freezes the strain state of the SrTiO₃ membranes, and is stable on the ~2 week time scale.

**Crystal structure characterization**. GIXRD measurements were performed using high-resolution X-ray diffractometer (PANalytical X'Pert Pro MRD). The measurement geometry was configured in the in-plane diffraction mode to allow X-rays to probe the crystal lattice along the in-plane directions.

**Domain structure characterization**. The PFM studies were carried out with a Cypher AFM (Asylum Research) using Ir/Pt-coated conductive tips (Nanosensor, PPP-EFM, force constant ≈2.8 N m⁻¹). The vector PFM mode was used to image both the out-of-plane and in-plane domain structure simultaneously.

**Optical second harmonic generation measurements**. In SHG measurements we illuminated the sample with a Coherent Chameleon Ultra II Ti: Sapphire laser tuned to 900 nm wavelength and focused onto the sample with a 20 × NA = 0.45 ELWD Nikon objective. The temperature-dependent SHG measurements were performed using a modified Janus ST500 Optical Vacuum Cryostat. Input polarization was controlled by the rotation of a half waveplate in the laser beam path. The generated SHG signals were collected on an Andor iXon CCD and Kymera Spectrometer.

The analysis of SHG intensity polar plots was performed using the symmetry-based SHG tensor. For the in-plane polarized domains, the fundamental beam produces electric field components which can be described as $E^\omega(\varphi) = (-E_0 sin\varphi, 0, E_0 cos\varphi)$, where $\varphi$ is the azimuthal angle of the fundamental light polarization shown in Fig. 3b. The light-induced non-linear polarization of the in-plane

polarized domains can be described using the *mm*2 point group-based SHG tensor:

$$\begin{pmatrix} P_1 \\ P_2 \\ P_3 \end{pmatrix} = \begin{pmatrix} 0 & 0 & 0 & 0 & d_{15} & 0 \\ 0 & 0 & 0 & d_{24} & 0 & 0 \\ d_{31} & d_{32} & d_{33} & 0 & 0 & 0 \end{pmatrix} \begin{pmatrix} E_1^2 \\ E_2^2 \\ E_3^2 \\ 2E_2E_3 \\ 2E_1E_3 \\ 2E_1E_2 \end{pmatrix}, \qquad (1)$$

where $E_1 = -E_0 sin\varphi$, $E_2 = 0$, and $E_3 = E_0 cos\varphi$. Since the output polarizer is placed either parallel ($I_x$, Fig. 3b) or perpendicular ($I_y$, Fig. 3b) to the in-plane uniaxial strain direction (polarization direction) in the membrane, the output SHG signal can be described as $I_x^{2\omega}(\varphi) = I_3^{2\omega} \propto (\mathbf{P}^{2\omega} \cdot \mathbf{A}_x)^2 = (d_{33}E_0^2 cos^2\varphi + d_{31}E_0^2 sin^2\varphi)^2$ and $I_y^{2\omega}(\varphi) = I_1^{2\omega} \propto (\mathbf{P}^{2\omega} \cdot \mathbf{A}_y)^2 = (d_{15}E_0^2 sin2\varphi)^2$, where $\mathbf{A}_x = (0, 0, 1)$ and $\mathbf{A}_y = (1, 0, 0)$. The experimental polar plots can be well fitted using the derived $I_x^{2\omega}(\varphi)$ and $I_y^{2\omega}(\varphi)$, indicating the strained SrTiO₃ is in the orthorhombic *mm*2 point group symmetry. Detailed fitting analysis can be found in ref. [33]. Note that the tetragonal 4*mm* point group-based SHG tensor also generates similar fitting results. But given the experimental strain geometry where the membrane elongates along only one in-plane direction while compresses along the out-of-plane direction due to the Poisson effect (we keep the membrane undeformed along the other orthogonal in-plane direction), the possibility of the tetragonal 4*mm* point group symmetry is thus ruled out from our analysis.

**Ginsburg–Landau–Devonshire calculations**. In this model, we used the power-series expansion of the Helmholtz free energy $F$ in terms of polarization components $P_i$ and structural order parameters $Q_i$ ($i = 1, 2, 3$), which is expressed as

follows[38]:

$$
\begin{aligned}
F ={}& \alpha_1\left(P_1^2 + P_2^2 + P_3^2\right) + \alpha_{11}\left(P_1^4 + P_2^4 + P_3^4\right) \\
&+ \alpha_{12}\left(P_1^2 P_2^2 + P_1^2 P_3^2 + P_2^2 P_3^2\right) + \beta_1\left(Q_1^2 + Q_2^2 + Q_3^2\right) \\
&+ \beta_{11}\left(Q_1^4 + Q_2^4 + Q_3^4\right) + \beta_{12}\left(Q_1^2 Q_2^2 + Q_1^2 Q_3^2 + Q_2^2 Q_3^2\right) \\
&+ 1/2c_{11}\left(S_1^2 + S_2^2 + S_3^2\right) + c_{12}\left(S_1 S_2 + S_1 S_3 + S_2 S_3\right) \\
&+ 1/2c_{44}\left(S_4^2 + S_5^2 + S_6^2\right) - g_{11}\left(S_1 P_1^2 + S_2 P_2^2 + S_3 P_3^2\right) \\
&- g_{12}\left[S_1\left(P_2^2 + P_3^2\right) + S_2\left(P_1^2 + P_3^2\right) + S_3\left(P_1^2 + P_2^2\right)\right] \\
&- g_{44}\left(S_4 P_2 P_3 + S_5 P_1 P_3 + S_6 P_1 P_2\right) - \lambda_{11}\left(S_1 Q_1^2 + S_2 Q_2^2 + S_3 Q_3^2\right) \\
&- \lambda_{12}\left[S_1\left(Q_2^2 + Q_3^2\right) + S_2\left(Q_1^2 + Q_3^2\right) + S_3\left(Q_1^2 + Q_2^2\right)\right] \\
&- \lambda_{44}\left(S_4 Q_2 Q_3 + S_5 Q_1 Q_3 + S_6 Q_1 Q_2\right) - t_{11}\left(P_1^2 Q_1^2 + P_2^2 Q_2^2 + P_3^2 Q_3^2\right) \\
&- t_{12}\left[P_1^2\left(Q_2^2 + Q_3^2\right) + P_2^2\left(Q_1^2 + Q_3^2\right) + P_3^2\left(Q_1^2 + Q_2^2\right)\right] \\
&- t_{44}\left(P_1 P_2 Q_1 Q_2 + P_1 P_3 Q_1 Q_3 + P_2 P_3 Q_2 Q_3\right),
\end{aligned}
\tag{2}
$$

where $S_n$ ($n = 1, 2, 3, 4, 5, 6$) are lattice strains, $c_{nl}$ are the elastic stiffnesses, $g_{nl}$ are the electrostrictive constants, $\lambda_{nl}$ are the linear-quadratic coupling coefficients between the strain and structural order parameters, and $t_{nl}$ are the coupling coefficients between the polarization and structural order parameters.

In order to simplify the analysis, we considered the scenario where there is no antiferrodistortive structural transition, i.e., $Q_i = 0$ ($i = 1, 2, 3$). In that care Eq. (2) can be written in the following format:

$$
\begin{aligned}
F ={}& \alpha_1\left(P_1^2 + P_2^2 + P_3^2\right) + \alpha_{11}\left(P_1^4 + P_2^4 + P_3^4\right) \\
&+ \alpha_{12}\left(P_1^2 P_2^2 + P_1^2 P_3^2 + P_2^2 P_3^2\right) + 1/2c_{11}\left(S_1^2 + S_2^2 + S_3^2\right) \\
&+ c_{12}\left(S_1 S_2 + S_1 S_3 + S_2 S_3\right) + 1/2c_{44}\left(S_4^2 + S_5^2 + S_6^2\right) \\
&- g_{11}\left(S_1 P_1^2 + S_2 P_2^2 + S_3 P_3^2\right) \\
&- g_{12}\left[S_1\left(P_2^2 + P_3^2\right) + S_2\left(P_1^2 + P_3^2\right) + S_3\left(P_1^2 + P_2^2\right)\right] \\
&- g_{44}\left(S_4 P_2 P_3 + S_5 P_1 P_3 + S_6 P_1 P_2\right).
\end{aligned}
\tag{3}
$$

Given the experimentally used in-plane uniaxial strain geometry, we considered the case of $S_1 \neq 0$, $S_2 = 0$, and $S_3 \neq 0$. In addition, in this case the shear strain $S_6$ is zero[38]. For the experimentally observed 180° ferroelectric domains, we can simplify the problem with the consideration of only one polarization component along the uniaxial strain direction, i.e., $P_1 \neq 0$, $P_2 = P_3 = 0$. Therefore, the Eq. (3) can be further simplified as:

$$
\begin{aligned}
F ={}& \alpha_1 P_1^2 + \alpha_{11} P_1^4 + 1/2c_{11}\left(S_1^2 + S_3^2\right) \\
&+ c_{12} S_1 S_3 + 1/2c_{44}\left(S_4^2 + S_5^2\right) \\
&- \left(g_{11} S_1 + g_{12} S_3\right)P_1^2.
\end{aligned}
\tag{4}
$$

Next, using the relation $\frac{\partial F}{\partial S_3} = \frac{\partial F}{\partial S_4} = \frac{\partial F}{\partial S_5} = 0$, which is derived from the mechanical boundary condition $\sigma_3 = \sigma_4 = \sigma_5 = 0$, we can derive the following relations:

$$
c_{11} S_3 + c_{12} S_1 - g_{12} P_1^2 = 0; \tag{5}
$$

$$
c_{44} S_4 = 0; \tag{6}
$$

$$
c_{44} S_5 = 0. \tag{7}
$$

From these relations, we can further derive $S_4 = S_5 = 0$ and $S_3 = \frac{g_{12} P_1^2 - c_{12} S_1}{c_{11}}$. Using these values in Eq. (4), we can rewrite the free energy in the following form with renormalized expansion coefficients:

$$
F = \alpha_{11}' P_1^4 + \alpha_1' P_1^2 + C, \tag{8}
$$

where $\alpha_{11}' = \alpha_{11} - \frac{g_{12}^2}{2c_{11}}$, $\alpha_1' = \alpha_1 - g_{11} S_1 + \frac{g_{12} c_{12} S_1}{c_{11}}$, $C = \frac{1}{2}c_{11} s_1^2 - \frac{c_{12}^2 S_1^2}{2c_{11}}$.

For the in-plane uniaxial strain geometry used in our experiment, $S_1 = S_u$ where $S_u$ is the uniaxial strain state. Using the thermodynamic parameters[38] (Supplementary Table 2) in Eq. (8), $F$ becomes a function of $P_1$, $S_u$, and temperature $T$ (since $\alpha_1$ relates to temperature). By calculating the minima of $F$ with respect to $P_1$, we can find the relation between $P_1$ and $T$ at a given $S_u$:

$$
P_1^2 = \frac{\left(g_{11} - \frac{g_{12} c_{12}}{c_{11}}\right)S_u - \alpha_1}{2\alpha_{11} - \frac{g_{12}^2}{c_{11}}}, \tag{9}
$$

where $\alpha_1 = 4.5\left[\coth\left(\frac{54}{T}\right) - \coth\left(\frac{54}{30}\right)\right] \times 10^{-3}$.

Such a relation is plotted in Supplementary Fig. 8, which allows us to extract $T_c$ at $P_1 = 0$ as a function of $S_u$ (Fig. 3d).

**First-principles density functional theory calculations.** First-principles DFT calculations were performed with QUANTUM-ESPRESSO[39] package using the LDA[40] and ultrasoft pseudopotentials from the Garrity, Bennet, Rabe, Vanderbilt high-throughput pseudopotential set[41]. We used an $8 \times 8 \times 8$ Monkhorst-Pack $k$-point mesh[42] for Brillouin-zone sampling and a plane-wave cutoff of 50 Ry and a charge density cutoff of 250 Ry. A force convergence threshold of $1.0 \times 10^{-4}$ y Bohr$^{-1}$, a pressure convergence threshold of 0.5 kbar, and Marzari–Vanderbilt smearing[43] of 1 mRy were used to optimize the atomic positions and lattice constants. In order to simplify the analysis, we use a 5-atom unit cell model, which rules out the possible effects of oxygen octahedral rotation, a known antiferroelectric mode in SrTiO$_3$. Our simplification is justified as the unstrained structure of interest in the experiment is cubic SrTiO$_3$ in which the oxygen octahedral rotation is zero above 105 K. The LDA lattice constant of cubic SrTiO$_3$ is 3.857 Å, which is consistent with previous theoretical results[44] and close to the experimental value of 3.905 Å. Following the experimental setup, the uniaxial strain state is induced by fixing the lattice constant $b$ along [010] to its ground-state value while fixing the lattice constant $a$ along [100] to a uniformly spaced grid sampled between 3.838 and 3.934 Å (corresponding to a strain range from −0.5% to 2.0% in reference to the DFT ground-state cubic structure) with a step size of 0.002 Å. For a given combination of ($a$, $b$), the perpendicular lattice constant $c$ and internal coordinates are fully relaxed to obtain the most stable structure under the given strain condition. The polarization is then estimated using the Born effective charges and the atomic positions. The detailed structural information can be found in the provided crystallographic information files (Supplementary Data 1).

**Molecular dynamics simulations.** MD simulations of SrTiO$_3$ were performed using a bond-valence-based atomistic potential parameterized from first-principles calculations[34,45]. This force field was able to reproduce both composition- and temperature-driven phase transitions of the Ba$_x$Sr$_{1-x}$TiO$_3$ solid solution[46] and has been successfully used to describe the THz-driven transient ferroelectric phase in SrTiO$_3$[47]. In order to model SrTiO$_3$ thin films without periodicity in the direction perpendicular to the surface, we constructed a supercell containing a SrTiO$_3$ slab of 90,000 atoms ($30 \times 20 \times 30$ cells) adjacent to a vacuum region. The SrTiO$_3$ slab, which is 20 unit cells in thickness, has the surface normal along the [010] direction. The vacuum along the surface normal is 85.4 Å thick, which is sufficiently large to prevent spurious interactions between neighboring periodic images. To stabilize the thin film in vacuum, the bond-valence charges of the surface layers were reduced by a factor of two, following a similar protocol developed in ref. [48]. In order to investigate the nature of temperature-driven phase transitions in strained SrTiO$_3$ membranes, we first equilibrated the slab model by running NVT (constant-volume constant-temperature) simulation using a time-step of 1 fs with a 2.0% tensile strain along the [100] direction. The temperature was controlled via the Nosé–Hoover thermostat[49] implemented in Large-scale Atomic/Molecular Massively Parallel Simulator[50]. To facilitate the equilibrium process, a small bias field was applied along the [100] direction for 50 ps after which another equilibrium run of 50 ps was performed with the bias field turned off. The final equilibrium structure is a single domain with the polarization along the [100] direction. Then we increased temperature gradually to explore the temperature-driven ferroelectric phase transition in strained SrTiO$_3$ membranes.

## Data availability
Crystallographic data in this study are provided in Supplementary Data 1. The data that support the findings of this study are available from the corresponding author upon reasonable request.

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

## Acknowledgements

We acknowledge fruitful discussions with Ian R. Fisher during the course of this work. This work was supported by the U.S. Department of Energy, Office of Basic Energy Sciences (DOE-BES), Division of Materials Sciences and Engineering, under contract no. DE-AC02-76SF00515. S.C. and D.L. were supported by the Gordon and Betty Moore Foundation's Emergent Phenomena in Quantum Systems Initiative through grant no. GBMF4415. V.H. was supported by the Air Force Office of Scientific Research (AFOSR) Hybrid Materials MURI under award no. FA9550-18-1-0480. R.X. also acknowledges partial support from Stanford Geballe Laboratory for Advanced Materials (GLAM) Postdoctoral Fellowship program. S. L. acknowledges support from the Westlake Foundation. Work at the Molecular Foundry was supported by the Office of Science, Office of Basic Energy Sciences, of the U.S. Department of Energy under Contract No. DE-AC02-05CH11231. Part of this work was performed at the Stanford Nano Shared Facilities (SNSF), supported by the National Science Foundation under award ECCS-1542152.

## Author contributions

R.X., H.Y.H., and D.L. conceived of the study. R.X., S.S.H., and S.C. designed the experiments. R.X. carried out the film synthesis, PFM characterization, and GLD calculations. J.H. and S.L. performed the DFT and MD simulations. R.X., E.S.B., E.K.W., and J.X. conducted the SHG measurements and analysis. R.X., T.J., V.H., and B.Y.W. performed the membrane fabrication. P.S., R.X., and S.S.H. carried out the GIXRD measurements. R.X., S.L., P.S., and H.Y.H. wrote the paper, with contributions from all authors.

## Competing interests

The authors declare no competing interests.
