## [Peer Review File · Nature Communications]

Reviewers' comments:

Reviewer #1 (Remarks to the Author):

In the manuscript by Xu et al., the authors report the observation of room-temperature ferroelectricity in uni-axial strained SrTiO₃ membranes. By SHG and PFM, the authors observed 180° ferroelectric domains at room temperature in SrTiO₃ membranes with large strain. Moreover, the authors also performed DFT and molecular dynamics simulations to explore the nature of the ferroelectric phase transition. Unlike the films on rigid substrates, this work demonstrates the continuous strain engineering of STO up to a record high value and explores the mechanism of intrinsic ferroelectricity in strained STO, which is of highly interest for the community. I would recommend publication after the authors address the following concerns/questions.

1. Currently, the strain is characterized by the displacement of the gold markers measured by optical microscopy. As the strain can be partially relaxed with microscopic cracks which may not be detected by optical microscopy or missed by microscopic PFM measurements. It is important to have a direct XRD characterization of the real strain and a,b,c lattice constants. Especially, the 2.5% strain is a record high value for SrTiO₃ and should be confirmed by direct measurements of the lattice constants.
2. How does the STO membranes start to form cracks as the strain increases? Is that a sudden or gradual change? It would be great if the authors can discuss how the ferroelectric domain changes beyond the maximum strain without forming cracks?
3. Are the measurements of different strained state performed on the same sample or on different samples?
4. There is a very relevant recent paper that demonstrated the giant uniaxial strain applied on freestanding PbTiO₃ membranes (Adv. Mater. Interfaces 1901604 (2020)).

Reviewer #2 (Remarks to the Author):

Using the PFM, SHG and first principles calculations, authors study the freestanding layers of SrTiO₃ and claim, "Room temperature robust ferroelectricity in strained STO". While PFM measurements and GLD calculations shows above room temperature FE, given the controversy surrounding PFM measurements and FE switching, I am afraid the claim of "robust FE above room temperature T_c" is rather weak. Other than that article is well written and overall a good read.

Few more technical comments:

Why induced FE is dynamic, does it decay as function of time?

How big is the tensile stress that is applied to keep the membrane undeformed along the other orthogonal in-plane direction; how the behavior changes if no such stress is applied.

Atoms does not have labeling in Fig 1.

Arrow in Fig 4c are too small and hard to see.

Reviewer #3 (Remarks to the Author):

The current manuscript reports the room-temperature ferroelectricity induced by uniaxial tensile strain in SrTiO₃, which is a typical centrosymmetric and paraelectric material. The epitaxial heterostructures of STO films with the Sr₂CaAl₂O₆ sacrificial buffer layer were grown to obtain the freestanding STO membranes. And by transferring the membranes onto a stretchable polymer sheet, a continuously tunable uniaxial strain could be applied. The in-plane ferroelectric polarizations along the tensile strain direction and the 180-degree domains were observed with the piezoresponse force microscopy, the symmetry lowering of the structure characterized with the second harmonic generation measurements, the energy stability of the strain-driven ferroelectric phase verified by the density functional theory calculations, while the nature of the ferroelectric phase transition was illustrated with molecular dynamics simulations. All these comprehensive investigations are solid and convincing. However, the presentation of the manuscript should be improved before being acceptable for publishing on Nature Communications.

1. It is an important method to apply the continuously tunable uniaxial strain to freestanding membranes and investigate the new behaviors being induced. To demonstrate its advantage, the unique response of STO should be clarified. For example, besides T_c, how do other ferroelectric characters (polarization strength, coercive field or domain structure) change upon the variation of strain? As a prospect, how would the anisotropy associated with the uniaxial strain be connected to potential applications?
2. The symmetry of the low-temperature ferroelectric phase was determined by SHG measurements. It would be more convincing if the results of other independent lattice measurements are given, e.g. the quantitative analyses of x-ray diffractions. By the way, the data are of high-quality – does this suggest the high quality of the interface between the STO membrane and the polyimide sheet? The authors are encouraged to comment on this.
3. The lattice structure of the low-temperature phase was determined by DFT calculations. The detail information, e.g. the cif file, should be presented and clearly described. And, how do the relaxed structures change with the uniaxial strain?
4. The discussions on Fig. 3b. What does “Distribution” refer to exactly? Under what strain the results were obtained – it should be specified (2% tensile strain along [100] direction) in the main text.
5. The MD results indicate the order-disorder character of the ferroelectric transition. There exist different polarization orientations as the temperature rising close to T_c. How can the MD results be correlated to the PFM and DFT results? How do the displacive and order-disorder mechanisms cooperate or compete?

Overall Response to Reviewers and Editors

We thank the Reviewers for their time in studying our manuscript and for the valuable comments that they have provided. Here, we provide responses that carefully address each individual comment. We have amended and improved our manuscript based on these comments and suggestions, and have also included additional editorial changes in our revised manuscript.

Response to Reviewer #1

Comment: “In the manuscript by Xu et al., the authors report the observation of room-temperature ferroelectricity in uni-axial strained SrTiO₃ membranes. By SHG and PFM, the authors observed 180° ferroelectric domains at room temperature in SrTiO₃ membranes with large strain. Moreover, the authors also performed DFT and molecular dynamics simulations to explore the nature of the ferroelectric phase transition. Unlike the films on rigid substrates, this work demonstrates the continuous strain engineering of STO up to a record high value and explores the mechanism of intrinsic ferroelectricity in strained STO, which is of highly interest for the community. I would recommend publication after the authors address the following concerns/questions.”

Response: We thank the Reviewer for the comprehensive summary and endorsement for publication.

Comment: “Currently, the strain is characterized by the displacement of the gold markers measured by optical microscopy. As the strain can be partially relaxed with microscopic cracks which may not be detected by optical microscopy or missed by microscopic PFM measurements. It is important to have a direct XRD characterization of the real strain and a,b,c lattice constants. Especially, the 2.5% strain is a record high value for SrTiO₃ and should be confirmed by direct measurements of the lattice constants.”

Response: We thank the Reviewer for this comment. Indeed, after submission, we also felt that this independent check would be valuable – not so much that we suspected cracks (since AFM characterization has shown that the strained membranes are intact and crack-free), but to rule out plastic deformation. Thus, we performed grazing incidence X-ray diffraction (GIXRD) measurements to probe the in-plane lattice parameters of strained membranes (Fig. R1). Here, with increasing strain, the GIXRD peak measured along the in-plane strain direction shifts towards lower angles (Fig. R1a), indicating the increase in the lattice parameter upon stretching, whereas the peak position measured along the in-plane unstrained direction remains almost unchanged, indicating the unstrained state along this direction (Fig. R1b). We also note that the uniaxial strain values measured by GIXRD closely match with the optically measured strain values (Fig. R1c). These measurement results further reveal and confirm the uniaxial strain geometry used in our experiment. These results are now included in the revised manuscript on page 4 and Fig. 2b-d.

Fig. R1 | Grazing incidence X-ray diffraction (GIXRD) measurement results. GIXRD results measured along **a**, [100] strain direction and **b**, [010] unstrained direction. **c**, Comparison of GIXRD strain and the optically measured nominal strain. The error bar is within the size of the marker.

Unfortunately, we were not able to complete the measurements up to 2.5% strain since our labs (and campus shared facilities where the XRD and PFM experiments were performed) were abruptly closed on Mar. 16. While the current schedule is uncertain, it is clear that our lab will be inaccessible for at least 1 month more, and the shared facilities beyond that. However, we believe the data we have in hand substantiate the central claim of the paper – namely, with XRD demonstrating up to 2% strain (with extrapolated T_c above 400 K), we can fully corroborate the primary claim of “robust room-temperature ferroelectricity in SrTiO₃”. To be conservative, we have now used values for 2% strain in the revised abstract, and data for 2% strain in the revised Fig. 2h, i and Fig. 3b. Given our observations of abrupt strain failure described below, giving us high confidence in the 2.5% data, we would also like to include the data point for 2.5% strain in the Supplementary Fig. 2 and 5 for completeness.

Comment: “How does the STO membranes start to form cracks as the strain increases? Is that a sudden or gradual change? It would be great if the authors can discuss how the ferroelectric domain changes beyond the maximum strain without forming cracks?”

Response: We observe abrupt crack formation in SrTiO₃ membranes beyond a maximum strain value (typically above 2.5% strain). Strain relaxation via the formation of cracks leads to a reduced T_c and paraelectricity in SrTiO₃ at room temperature. Thus, we were not

Fig. R2 | Piezoresponse Force Microscopy (PFM) characterization of strained SrTiO₃ membranes beyond the threshold for crack formation. **a**, PFM height image shows cracks perpendicular to the uniaxial strain direction. White arrows here illustrate the uniaxial strain direction along [100]. **b**, Lateral phase and **c**, amplitude images show the absence of 180° degree domains in cracked SrTiO₃ membranes.

able to probe the ferroelectric domains at room temperature in SrTiO₃ membranes beyond the point of crack formation (Fig. R2). To clarify these points, we have added a discussion in the revised manuscript on page 5-6 and Supplementary Fig. 4.

Comment: “Are the measurements of different strained state performed on the sample sample or on different samples?”

Response: The PFM measurements presented were performed on the same sample as a function of strain. For SHG measurements, which were performed at Lawrence Berkeley National Lab, we prepared and measured multiple samples with different strain states, in order to better utilize the user facility time (changing the strain state could only be performed in our experimental setup at Stanford). In all cases, no notable difference was observed in all measurements for different samples at the same strain state.

Comment: “There is a very relevant recent paper that demonstrated the giant uniaxial strain applied on freestanding PbTiO₃ membranes (Adv. Mater. Interfaces 1901604 (2020)).”

Response: We thank the Reviewer for pointing out this relevant reference, which is now cited in the revised manuscript on page 3.

Response to Reviewer #2

Comment: “Using the PFM, SHG and first principles calculations, authors study the freestanding layers of SrTiO₃ and claim, “Room temperature robust ferroelectricity in strained STO”. While PFM measurements and GLD calculations shows above room temperature FE, given the controversy surrounding PFM measurements and FE switching, I am afraid the claim of “robust FE above room temperature T_c” is rather weak. Other than that article is well written and overall a good read.”

Response: We thank the Reviewer for the positive appraisal of the article. We agree that there are other mechanisms and artifacts that could lead to ferroelectric-like PFM responses, such as electrostriction caused by charge injection, electrochemical strain due to ionic migration, and shear deformation due to surface contamination or topographical features, etc¹. Indeed, these factors pose challenges to the correct interpretation of PFM results for perpendicularly polarized states. However, the *lateral* PFM results we provided in our manuscript, showing 180° polydomain patterns with clear phase contrast, evidenced the presence of ferroelectricity without ambiguity. First, we carefully studied the polarization direction of the domains by aligning the PFM cantilever along different scan directions (Supplementary Fig. 3). These results clearly show that the polarization of these domain features align along specific crystallographic directions, whereas other mechanisms of induced ferroelectric-like response are typically invariant with respect to the cantilever scan orientation. Second, the domain signals we probed are robust and stable which do not decay with time (so long as the strain state is maintained). Third, there are no similar topographical features on the membrane surface that resemble these domain

features, which rules out the possibility of crosstalk signals caused by surface contamination or topographical features. To clarify these points, we have added a discussion in the revised caption of Supplementary Fig. 3.

In addition, in our studies the PFM results are not the only evidence to corroborate the presence of room-temperature ferroelectricity in strained SrTiO₃ membranes. The SHG measurements, which clearly show the polar symmetry in strained membranes at room temperature, again demonstrate the intrinsic structural origin of the observed ferroelectricity. Note that previous work has primarily utilized and relied on SHG to probe ferroelectricity driven by lattice mismatch strain in SrTiO₃ thin films².

Comment: “Few more technical comments: Why induced FE is dynamic, does it decay as function of time?”

Response: The induced ferroelectricity is stable and does not decay while the strain state is maintained (and it is stable at least on the several-week timescale in our measurements). When the strain is released, then ferroelectricity disappears. Compared to the film grown on a rigid substrate with fixed strain, the strain in our membrane setup can be released by melting the polymer adhesive, which allows us to re-stretch the membrane into new strain states. Our experiment utilized such dynamically tunable strain to induce and control ferroelectricity in SrTiO₃.

Comment: “How big is the tensile stress that is applied to keep the membrane undeformed along the other orthogonal in-plane direction; how the behavior changes if no such stress is applied.”

Response: Since stress was not directly measured in our experiment (in our work we measured strain either optically or using GIXRD), we can estimate the stress values based on the simulation results from finite element analysis. In this simulation, we applied the displacement boundary conditions on the vertical facets of the polyimide substrate. For the simulation of the uniaxial strain field geometry, the displacement is only allowed along the

Fig. R3 | Finite element analysis of stress and strain distribution simulations. **a**, stress and **b**, strain distribution results for SrTiO₃ and the supporting polyimide sheet under the uniaxial strain condition.

[100] direction. Using the bulk values of Young's modulus of the oxide layer and the polyimide substrate, we simulated both the stress and strain distributions (Fig. R3). It is noted that ~ 0.017 GPa stress is required to keep the membrane undeformed along the orthogonal in-plane direction when the uniaxial strain along the [100] direction is 2%.

Qualitatively, the behavior is similar to the case without this compensating orthogonal stress. We chose to fix the orthogonal lattice parameters to the unstrained state mainly for the following two reasons: 1. To have a well-controlled uniaxial strain geometry for quantitative comparison to the computational studies; 2. Minimize the potential for the formation of wrinkles along the orthogonal direction since compressive stress often induces wrinkling in thin membrane sheets.

Comment: "Atoms does not have labeling in Fig 1."

Response: We thank the Reviewer for pointing this out, and we have revised Fig. 1 to include the additional labeling.

Comment: "Arrow in Fig 4c are too small and hard to see."

Response: We thank the Reviewer for noting this, and we have revised Fig. 4c to make the arrows more visible.

Response to Reviewer #3

Comment: "The current manuscript reports the room-temperature ferroelectricity induced by uniaxial tensile strain in SrTiO₃, which is a typical centrosymmetric and paraelectric material. The epitaxial heterostructures of STO films with the Sr₂CaAl₂O₆ sacrificial buffer layer were grown to obtain the freestanding STO membranes. And by transferring the membranes onto a stretchable polymer sheet, a continuously tunable uniaxial strain could be applied. The in-plane ferroelectric polarizations along the tensile strain direction and the 180-degree domains were observed with the piezoresponse force microscopy, the symmetry lowering of the structure characterized with the second harmonic generation measurements, the energy stability of the strain-driven ferroelectric phase verified by the density functional theory calculations, while the nature of the ferroelectric phase transition was illustrated with molecular dynamics simulations. All these comprehensive investigations are solid and convincing. However, the presentation of the manuscript should be improved before being acceptable for publishing on Nature Communications."

Response: We thank the Reviewer for the comprehensive summary, positive endorsement, and suggestions for improvement which we address below.

Comment: "It is an important method to apply the continuously tunable uniaxial strain to freestanding membranes and investigate the new behaviors being induced. To demonstrate its advantage, the unique response of STO should be clarified. For example, besides T_c, how do other ferroelectric characters (polarization strength, coercive field or domain

structure) change upon the variation of strain? As a prospect, how would the anisotropy associated with the uniaxial strain be connected to potential applications?”

Response: One unique aspect of our work is the ability to artificially create in-plane polarized 180° domain structures in SrTiO_3 with uniaxial strain geometry. First, previous work has primarily focused on the case of biaxial strain in thin film SrTiO_3 , where a polydomain structure involving multiple polarization variants has been revealed by phase-field simulations^{3,4}. However, it is also noted that very little experimental work has been done so far to directly study the ferroelectric domain structure in SrTiO_3 films via PFM. This is mostly because of the limited strain achieved using rigid substrates (before defect formation becomes important), leading to a weak piezoelectric response at room temperature that is not sufficient to allow clear PFM evidence. Thus, one advantage of our work is the capability to create enhanced polarization and piezoelectric response to enable the study of domain and domain wall structures in ferroelectric SrTiO_3 . In addition, the observed in-plane polarized 180° domain structure with only one in-plane polarization variant (along the $[100]/[\bar{1}00]$ directions) is rather rare in epitaxial thin films, not just in SrTiO_3 but also in other ferroelectric materials such as PbTiO_3 , BaTiO_3 , etc. To the best of our knowledge, the only other example we can find is the recent work which reports a 180° stripe domain pattern with polarizations along the $[111]/[1\bar{1}\bar{1}]$ directions in BiFeO_3 thin films grown on GdScO_3 (010) substrates⁵. Thus, the ability of our work to artificially create such domain structures provides the opportunities for exploring and controlling new functionalities in ferroelectrics, which could become potential candidates for device elements in next-generation nanoelectronics such as in-plane ferroelectric non-volatile memories. We have clarified these points and added a discussion in the revised manuscript on page 5.

In our work we use SrTiO_3 membranes as an example to demonstrate the effects of uniaxial strain on ferroelectricity in SrTiO_3 , but looking forward, there are more possibilities in future to utilize the degree of freedom of this flexible strain platform to manipulate and control domain structures in other ferroelectric materials. The ability to create arbitrary anisotropic strain fields could allow the control of further possible non-trivial domain structures and topologies, bringing in new functionalities for potential device applications.

Comment: “The symmetry of the low-temperature ferroelectric phase was determined by SHG measurements. It would be more convincing if the results of other independent lattice measurements are given, e.g. the quantitative analyses of x-ray diffractions.”

Response: We thank the Reviewer for this comment. Indeed, after submission, we also felt that this independent check would be valuable – not so much that we suspected cracks (since AFM characterization has shown that the strained membranes are intact and crack-free), but to rule out plastic deformation. Thus, we performed grazing incidence X-ray diffraction (GIXRD) measurements to probe the in-plane lattice parameters of strained membranes (Fig. R4). Here, with increasing strain, the GIXRD peak measured along the in-plane strain direction shifts towards lower angles (Fig. R4a), indicating the increase in the lattice parameter upon stretching, whereas the peak position measured along the in-plane unstrained direction remains almost unchanged, indicating the unstrained state along

this direction (Fig. R4b). We also note that the uniaxial strain values measured by GIXRD closely match with the optically measured strain values (Fig. R4c). These measurement results further reveal and confirm the uniaxial strain geometry used in our experiment. These results are now included in the revised manuscript on page 4 and Fig. 2b-d.

Unfortunately, we were not able to complete the measurements up to 2.5% strain since our labs (and campus shared facilities where the XRD and PFM experiments were performed) were abruptly closed on Mar. 16. While the current schedule is uncertain, it is clear that our lab will be inaccessible for at least 1 month more, and the shared facilities beyond that. However, we believe the data we have in hand substantiate the central claim of the paper – namely, with XRD demonstrating up to 2% strain (with extrapolated T_c above 400 K), we can fully corroborate the primary claim of “robust room-temperature ferroelectricity in SrTiO₃”. To be conservative, we have now used values for 2% strain in the revised abstract, and data for 2% strain in the revised Fig. 2h, i and Fig. 3b. Given our observations of abrupt strain failure described below, giving us high confidence in the 2.5% data, we would also like to include the data point for 2.5% strain in the Supplementary Fig. 2 and 5 for completeness.

Fig. R4 | Grazing incidence X-ray diffraction (GIXRD) measurement results. GIXRD results measured along **a**, [100] strain direction and **b**, [010] unstrained direction. **c**, Comparison of GIXRD strain and the optically measured nominal strain. The error bar is within the size of the marker.

Comment: “By the way, the data are of high-quality – does this suggest the high quality of the interface between the STO membrane and the polyimide sheet? The authors are encouraged to comment on this.”

Response: We think the following factors contribute to the quality of our SHG data: first, the measurement optics are well aligned ensuring a good signal-to-noise ratio. Second, in SHG measurements, the incident beam is polarized within the plane parallel to the membrane surface. Such an optical geometry is sensitive to the symmetry of in-plane ferroelectric polarization. Since the tensile strain induced polarization is purely in-plane oriented, this can effectively maximize the polarization component detected by SHG. Perhaps most importantly, as the Reviewer notes, and now included in the revised text on page 3, the interface between the STO membrane and the polyimide sheet is central to providing strong interface adhesion, enabling the success of the strain experiment. Clean and flat polyimide surfaces give rise to better adhesion and interfacial bonding with STO membranes, which also reduces crack formation upon stretching.

Comment: “The lattice structure of the low-temperature phase was determined by DFT calculations. The detail information, e.g. the cif file, should be presented and clearly described. And, how do the relaxed structures change with the uniaxial strain?”

Response: Our DFT calculations reveal that the a and c lattice parameters of SrTiO₃ increase and decrease respectively with the uniaxial strain, coupled with the local displacements of Ti and Sr atoms along the [100] direction (Fig. R5). In response to the uniaxial strain along the [100] direction, both the Sr and Ti atoms displace with respect to the surrounding oxygen lattice, therefore both contributing to the total polarization. This structural information is now included in Supplementary Fig. 9 and Table 1. In addition, all the CIF files are now provided in Supplementary Data.

Fig. R5 | Calculated local displacements of Sr and Ti atoms along the [100] direction as a function of [100] uniaxial strain in SrTiO₃ from DFT calculations.

Comment: “The discussions on Fig. 3b. What does “Distribution” refer to exactly? Under what strain the results were obtained – it should be specified (2% tensile strain along [100] direction) in the main text.”

Response: We thank the Reviewer for the careful reading. Here the “Distribution” refers to the probability distribution of the unit cells adopting a specific local polarization. Such a distribution analysis has been used to characterize the nature of the phase transition of various ferroelectric materials such as PbTiO₃^{6,7}, BiFeO₃⁸, and BaTiO₃⁹. We have clarified this point in the revised text on page 7 and the caption of Fig. 4. We have also included the strain condition for MD simulations in both the main text on page 7 and the caption of Fig. 4.

Comment: “The MD results indicate the order-disorder character of the ferroelectric transition. There exist different polarization orientations as the temperature rising close to T_c. How can the MD results be correlated to the PFM and DFT results? How do the displacive and order-disorder mechanisms cooperate or compete?”

Response: First, the correlation between the MD simulations and the experiment is 1) to demonstrate computationally that strained SrTiO₃ exhibits a net polar-nonpolar phase

transition, i.e. a ferroelectric transition, and 2) to reveal the atomistic character of the nature of the phase transition, which is difficult to obtain experimentally. The MD results showing the presence of an order-disorder character also agree with the implications from DFT calculations. As shown in Fig. 4a, the energy difference between the polar and non-polar phase calculated by DFT remains relatively small, therefore upon thermal activation, atoms can relatively easily overcome the barrier between different polar states, giving rise to the order-disorder transition behavior.

Regarding the second question, we find that at lower temperature the displacive character is the main feature, as the distribution of local polarization has a single peak and the peak position shifts toward lower polarization values with increasing temperature. When the temperature rises close to T_c , the order-disorder character dominates, shown as a double-peak on the distribution curve. However, an ideal order-disorder character will have the two peaks well separated in the high temperature paraelectric phase (Fig. R6). Therefore, even at the temperature close to T_c , the displacive character still plays a role, indicating a mixture of displacive and order-disorder transition characteristics in SrTiO₃. We have now included these discussions in Supplementary Fig. 10 for clarity.

Fig. R6 | Schematics of the polarization distribution in paraelectric phases near a displacive transition, order-disorder transition, and a mixture of the two.

References

1. Gruverman, A., Alexe, M. & Meier, D. Piezoresponse force microscopy and nanoferroic phenomena. *Nat. Commun.* **10**, 1661 (2019).
2. Haeni, J. H. *et al.* Room-temperature ferroelectricity in strained SrTiO₃. *Nature* **430**, 758–761 (2004).
3. Li, Y. L. *et al.* Phase transitions and domain structures in strained pseudocubic (100) SrTiO₃ thin films. *Phys. Rev. B* **73**, 1–13 (2006).
4. Chen, L. Q. *et al.* Phase transitions and domain stabilities in biaxially strained (001) SrTiO₃ epitaxial thin films. *J. Appl. Phys.* **108**, 084113 (2010).
5. Chen, Z. *et al.* 180° ferroelectric stripe nanodomains in BiFeO₃ thin films. *Nano Lett.* **15**, 6506–6513 (2015).
6. Shin, Y. H., Cooper, V. R., Grinberg, I. & Rappe, A. M. Development of a bond-valence molecular-dynamics model for complex oxides. *Phys. Rev. B* **71**, 054104 (2005).
7. Shin, Y. H., Son, J. Y., Lee, B. J., Grinberg, I. & Rappe, A. M. Order-disorder character of PbTiO₃. *J. Phys. Condens. Matter* **20**, 015224 (2008).
8. Liu, S., Grinberg, I. & Rappe, A. M. Development of a bond-valence based interatomic potential for BiFeO₃ for accurate molecular dynamics simulations. *J. Phys. Condens. Matter* **25**, 102202 (2013).
9. Qi, Y., Liu, S., Grinberg, I. & Rappe, A. M. Atomistic description for temperature-driven phase transitions in BaTiO₃. *Phys. Rev. B* **94**, 134308 (2016).

REVIEWER COMMENTS

Reviewer #1 (Remarks to the Author):

In the revised version of the manuscript, the authors have addressed all my concerns properly and I would highly recommend publication in Nature Communications.

Reviewer #2 (Remarks to the Author):

Authors have answered my questions satisfactorily.

Additionally, I would like authors to explain how the value of α_{11}^{\prime} , α_1^{\prime} and C in equation (3) is derived from (2) explicitly, if require show steps in the supplementary materials. Seems like authors have not used these parameters (g_{12} and C_{12}) anyway.

Right below, equation 3, I believe the value (expression involving cotangent) belongs to α_1^{\prime} ; please correct it. Please include the used value of α_{11}^{\prime} .

Reviewer #3 (Remarks to the Author):

The authors replied to all of my questions carefully and addressed the according issues in the revised manuscript. I recommend accepting the paper for publication.

Response to Reviewer #1

Comment: “In the revised version of the manuscript, the authors have addressed all my concerns properly and I would highly recommend publication in Nature Communications.”

Response: We thank the Reviewer for the recommendation for publication.

Response to Reviewer #2

Comment: “Authors have answered my questions satisfactorily.”

Response: We thank the Reviewer for the positive endorsement.

Comment: “Additionally, I would like authors to explain how the value of α_{11}^{\prime} , α_1^{\prime} and C in equation (3) is derived from (2) explicitly, if require show steps in the supplementary materials. Seems like authors have not used these parameters (g_{12} and C_{12}) anyway.”

Response: In the revised manuscript, we include the detailed, step-by-step derivation that explicitly explains how these renormalized expansion coefficients (α_{11}^{\prime} , α_1^{\prime} , and C) are derived from our calculation. These details are now highlighted in the revised Methods section on page 13 – 14. In our modeling, g_{12} and c_{12} were used for the calculation of P_1 , i.e., $P_1^2 = \frac{(g_{11} - \frac{g_{12}}{c_{11}})S_u - \alpha_1}{2\alpha_{11} - \frac{g_{12}^2}{c_{11}}}$, and the related derivation process and details are now included in the revised manuscript on page 14.

Comment: “Right below, equation 3, I believe the value (expression involving cotangent) belongs to α_1^{\prime} ; please correct it. Please include the used value of α_{11}^{\prime} .”

Response: We respectfully disagree with the Reviewer on this point. The value of $4.5[\coth(\frac{54}{T}) - \coth(\frac{54}{30})] \times 10^{-3}$ belongs to α_1 . α_1 is the only coefficient in our calculation which relates to temperature. The value of α_1 for SrTiO₃ can be found from literature (N. A. Pertsev *et al. Phys. Rev. B* **61**, R825-829 (2000)). For clarity, we now include the value of all the Landau coefficients of SrTiO₃ in Supplementary Table 2.

Response to Reviewer #3

Comment: “The authors replied to all of my questions carefully and addressed the according issues in the revised manuscript. I recommend accepting the paper for publication.”

Response: We thank the Reviewer for the recommendation for publication.

REVIEWERS' COMMENTS:

Reviewer #2 (Remarks to the Author):

I commend authors for providing details of derivation according to my question. I am satisfied with their answer; the paper can be published.

I notice possible typos in equations 7 (possibly c_{44} instead of c_{11}) and equation 9 (possibly g_{12} c_{12}/c_{11} instead of g_{12}/c_{11}). I trust authors to make appropriate changes; no additional review is required, in my opinion.

Response to Reviewer #2

Comment: I commend authors for providing details of derivation according to my question. I am satisfied with their answer; the paper can be published.

Response: We thank the Reviewer for the recommendation for publication.

Comment: I notice possible typos in equations 7 (possibly c_{44} instead of c_{11}) and equation 9 (possibly $g_{12} c_{12}/c_{11}$ instead of g_{12}/c_{11}). I trust authors to make appropriate changes; no additional review is required, in my opinion.

Response: We apologize for these typos and thank the Reviewer for the detailed reading. We have corrected these typos in the revised manuscript.